# Microstructure and Abrasive Wear Resistance of Various Alloy Hardfacings for Application on Heavy-Duty Chipper Tools in Forestry Shredding and Mulching Operations

**DOI:** 10.3390/ma12132212

**Published:** 2019-07-09

**Authors:** Ladislav Falat, Miroslav Džupon, Miroslava Ťavodová, Richard Hnilica, Veronika Ľuptáčiková, Lucia Čiripová, Viera Homolová, Katarína Ďurišinová

**Affiliations:** 1Institute of Materials Research, Slovak Academy of Sciences, Watsonova 47, 04001 Košice, Slovakia; 2Faculty of Environmental and Manufacturing Technology, Technical University in Zvolen, T. G. Masaryka 24, 96001 Zvolen, Slovakia

**Keywords:** forestry chipper tool, hard surfacing, microstructure, hardness, abrasive wear

## Abstract

Five different alloy hardfacings on 16MnCr5 grade low-carbon ferritic–pearlitic steel were investigated in terms of their abrasive wear resistance in laboratory testing conditions. The selected hardfacing materials, namely “E520 RB”, “RD 571”, “LNM 420FM”, “E DUR 600”, and “Weartrode 62”, were individually deposited onto plain ground-finish surfaces of 10 mm thick steel plate samples. The studied hardfacings were fabricated using several different welding methods and process parameters proposed by their industrial manufacturers. In the present comparative study, the results obtained from laboratory abrasive wear tests of the investigated hardfacings were analyzed and discussed in relation to their microstructure, hardness, and wear mechanism characteristics. Regardless of great variety in microstructure and chemical composition of individual hardfacing materials, the results clearly indicated the governing factor for the wear resistance improvement to be the overall carbon content of the used hardfacing material. Thus it has been shown that the “E520 RB” hardfacing exhibited the highest abrasive wear resistance thanks to its appropriate hardness and beneficial “ledeburite-type” eutectic microstructure.

## 1. Introduction

In the frame of the worldwide trend of environmental protection in forestry and sustainable development of forest environment, it has been become an important issue to address the possibilities of effective exploitation of natural resources by utilization of modern, highly efficient and safe forestry maintenance equipment [1]. A substantial task in forestry is related to land clearing, i.e., regular removal of undesired (overgrowth) woody vegetation, e.g., in landscape areas intended for new forest cultivation, the areas alongside forest roads, and more importantly the areas below overhead power lines. The permanent requirement for lifetime increase of working tools of forestry maintenance equipment, such as shredders and/or mulchers, represents a strong impulse for seeking economic and efficient material/technological solutions aiming at significant suppression of mechanical degradation of the tools’ functional surfaces [2]. In general, the operation of chipper tools of forestry machines is characterized by great mechanical loading in a highly heterogeneous work environment. This environment includes inhomogeneous wood mass and soil containing rocks and minerals of various size, hardness and distribution. Therefore, the tools have to be made of materials with appropriate strength, toughness, and fatigue resistance, i.e., the resistance against brittle fracture during cyclic loading, and abrasive wear resistance. At the same time, the tools’ material and production costs have to meet satisfactory economic criteria. The possibilities for lifetime improvement of the tools are of primary interest for end-users implementing the forestry maintenance in practice. The main problem to be solved is related to effective protection of the tools’ functional parts against accelerated damage and failure in service conditions. Since one of the main degradation mechanisms of the tools’ functional surfaces is considered abrasive wear, adequate measures are to be undertaken for its suppression [2,3]. To fulfil such a requirement, several actions can be considered realistic. Commercially, hard soldering of tungsten–carbide tips on the functional surface of the tool represents the most frequently utilized method. Cemented carbides, or hardmetals, are widely used for extremely demanding applications due to their hardness and strength, fracture toughness, and excellent wear resistance [4]. Another possible measures that may be taken into consideration, are the following: the whole tool quenching-and-tempering treatment, its surface carburizing, surface hardening, superficial alloying by laser or electric arc remelting, and hard surfacing [5,6,7,8,9,10,11,12,13]. According to current in-service experience, premature failures of chipper tools still remain a pertinent issue in forestry which may be a consequence of the fact that degradation behavior of forestry tools has not been up to now so widely investigated, compared to other heavy-duty tools employed in, e.g., agriculture, road building, and the mining industry [14]. Taking into account a great complexity of the tools loading conditions, there is no trivial way to assure their desirable durability during operation. Therefore, each of the above proposed remedial actions for the tools’ protection against abrasive wear, has to be investigated separately. In our former study [15], the effectiveness of quenching-and-tempering heat treatment of 16MnCr5 grade low-carbon steel was investigated in terms of its effect on abrasive wear resistance in relation with microstructure, hardness, and notch toughness characteristics. It has been concluded that the used heat treatment resulted in a certain recognizable improvement of abrasive wear resistance as a result of microstructural modification from ferritic-pearlitic microstructure in initial as-received material state to a more beneficial tempered martensitic microstructure in the heat-treated material state. At the same time, the quenched-and-tempered samples exhibited higher values of hardness and notch toughness compared to the untreated ones. However, the obtained improvement of the properties due to the application of heat treatment was relatively small. When selecting appropriate material for working in abrasive wear conditions, the criteria generally considered are the material hardness and microstructure [16,17,18,19,20,21]. However, there is no unified opinion among researchers on the most beneficial type of material microstructure in terms of its abrasive wear resistance. Depending on specific application, either martensitic-carbidic or austenitic-carbidic microstructures are considered to be the most suitable for abrasive wear performance [22,23,24]. These different views arise from the fact of great diversity in abrasive wear processes as well as the wide range of tool operational conditions. Therefore, it is highly important that materials investigations aimed at improvement of abrasive wear resistance of specific tools or structures always take into account the mentioned considerations.

In the present study, five different alloy hardfacings on 16MnCr5 grade low-carbon ferritic-pearlitic steel are investigated in terms of their abrasive wear resistance in laboratory testing conditions. The observed differences in wear resistance among individual hardfacing materials are discussed in relation to their microstructures, hardness, and wear mechanism characteristics.

## 2. Materials and Methods

The low carbon 16MnCr5 grade ferritic-pearlitic steel, representing a common material for industrial production of forestry chipper tools, was used in the present study as a substrate reference material for hard surfacing. Its chemical composition is shown in Table 1. In experimental work performed within the present investigation, all the testing specimens produced from this reference material were assigned number “0”. Altogether, five different commercial alloy hardfacings were individually deposited onto plain “ground finish” surfaces of 10 mm thick steel plate samples. The producer of the original hardfacing materials for the preparation of “E520 RB” and “RD 571” hardfacings is Welding Research Institute (Výskumný ústav zváračský—z. z. p. o.), Bratislava, Slovakia.

The electrode supplier for “LNM 420FM” hardfacing is Lincoln Electric Europe, Praha, Czech Republic. The “E DUR 600” electrode manufacturer is Elektrode Jesenice d.o.o., Slovenia. The “Weartrode 62” electrode supplier is ESAB Slovakia s.r.o., Bratislava, Slovakia. All the hardfacings were fabricated as two-layer deposits. The chemical composition of all hardfacing materials, used for fabrication of individual hardfacing deposits on 16MnCr5 grade steel substrate, is listed in Table 2.

In all further experimental work, the specimens referring to individual hardfacings have been numbered successively, according to their listing in Table 2. The selection of particular hardfacing materials was based on the assumption of creation of substantially differing microstructures and phase compositions of individual hardfacings in order to perform their pilot comparative study in terms of their abrasive wear resistance in laboratory testing conditions. The studied hardfacings were fabricated using several different welding methods and process parameters, recommended by their industrial manufacturers (see Table 3).

Microstructural observations of the produced hardfacings were performed using light-optical microscope (LOM) OLYMPUS GX71 (OLYMPUS Europa Holding GmbH, Hamburg, Germany) and scanning electron microscope (SEM) JEOL JSM-7000F (Jeol Ltd., Tokyo, Japan) equipped with energy-dispersive X-ray (EDX) analyzer INCA X-sight model 7557 (Oxford Instruments, Abingdon, Oxfordshire, UK). Predicted equilibrium phase composition of the hardfacings has been calculated using thermodynamic software Thermo-Calc (Thermo-Calc Software AB, Solna, Sweden) employing thermodynamic database TCFE6. Experimental study of phase composition of the hardfacings was studied by X-ray diffraction (XRD) method. The XRD measurements were carried out on a Philips X’Pert Pro diffractometer (Panalytical B.V., Almelo, The Netherlands) in Bragg–Brentano geometry, using Cu–Kα radiation and ultra-high-speed detector X’Celerator. In order to characterize surface hardness of individual hardfacings, conventional Vickers hardness measurements were carried out using Vickers 432 SVD hardness tester (Wolpert Wilson Instruments, division of Instron Deutschland GmbH, Aachen, Germany), applying test conditions for HV10 unit scale (i.e., load of 98 N, dwell time 10 s). Three-body abrasion wear tests were performed on an experimental testing apparatus shown in Figure 1, operating at testing conditions in conformity with “ГОСТ 23.208-79” Russian standard [25].

The dimensions of plain “ground finish” samples for abrasive wear tests were 30 × 30 × 7 mm. The abrasion wear test procedure included the sample fixation within the testing machine, an initiation of SiO_2_ abrasive feeding from the feed hopper onto the sample surface and simultaneous pressing the tested sample with the abrasive against rotary rubber cylinder with a rotary speed of 1000 rpm. The sliding distance and applied force on the sample during abrasion wear test were 767.75 m and 15.48 N, respectively. For each tested material, three test samples were examined. Each sample was twice weighted using an analytical balance, KERN ABJ 120-4 M (KERN & SOHN GmbH, Balingen, Germany) before and after the abrasion wear test. The calculated average weight losses of the tested specimens were obtained from five consecutive measurements per individual specimen and were considered to be the criterion for abrasive wear resistance. Apart from individual hardfacings, an original 16MnCr5 steel material was also tested for reference. Thus, a relative abrasive wear resistance (Ψ_abr._) of individual hardfacings could also be determined:Ψ_abr._ = Δm_ref._/Δm_spec._(1)
Δm_ref._—average weight loss of reference (16MnCr5 steel) material (g)Δm_spec._—average weight loss of specimen (hardfacing) material (g)

The obtained Ψ_abr._ values were graphically correlated with corresponding hardness ratio K_T_:K_T_ = H_mater._/H_abr._(2)
H_mater._—average hardness of tested material (a.u.)H_abr._—average hardness of used abrading agent (a.u.)


In the present case, 96% purity silica (SiO_2_) sand abrading agent was used. Its granularity was in the range from 0.1 mm to 0.3 mm and its humidity was about 1%. Its average hardness was 54 HRC corresponding to 576 HV10. The individual K_T_ ratios were calculated using average Vickers hardness values of the hardfacings surfaces. The average hardness values were calculated from ten independent surface hardness measurements. After the abrasion wear tests, depth profiles of individual wear tracks were characterized by optical 3D profiler PLu neox (Sensofar, Barcelona, Spain). The corresponding wear mechanisms were analysed by scanning electron microscope (SEM) JEOL JSM-7000F (Jeol Ltd., Tokyo, Japan) using secondary electrons imaging (SEI) contrast.

## 3. Results and Discussion

### 3.1. Microstructure

Figure 2, Figure 3, Figure 4, Figure 5 and Figure 6 show heterogeneous microstructures and corresponding phase analyses of the individual hardfacings deposited onto 16MnCr5 grade low-carbon steel substrate. Figure 2a represents light-optical micrograph of cast dendritic microstructure of “E520 RB” hardfacing produced by MMA welding method. The more detailed SEM microstructure is shown in Figure 2b. Thanks to the “E520 RB” hardfacing chemical composition (Table 2), including extremely high chromium and carbon content (25 wt.% Cr, 3.5 wt.% C), the microstructure is formed of a coarse eutectic “ledeburite-type” structure containing primary δ-ferrite dendrites and interdendritic carbides. These carbides were predicted by thermodynamic calculation (Figure 2c) to be the chromium carbides of M_7_C_3_ type, showing subsequent on-cooling phase transformation to M_23_C_6_ carbide type. However, it should be noted that due to fast solidification kinetics during the hardfacing preparation, the solidified hardfacing microstructure is not in equilibrium state. Thus, the occurrence of both higher-temperature M_7_C_3_ and lower-temperature M_23_C_6_ carbides may be expected in the microstructure. For the same reason, the precipitation of the predicted intermetallic FeCr-based σ-phase (Figure 2c) is not very realistic within the as-cast hardfacing microstructure. A corresponding XRD pattern of the hardfacing is shown in Figure 2d. It can be clearly seen that performed XRD analysis confirmed the presence of primary ferrite phase and eutectic carbide of M_7_C_3_ type.

Figure 3a shows the light-optical micrograph of “cermet-type” microstructure of “RD 571” hardfacing produced by autogenous welding using a cored electrode with W_2_C filler powder. Due to the “RD 571” hardfacing chemical composition (Table 2), including extremely high nickel content (15 wt.% Ni), the matrix phase of this hardfacing is primarily formed of austenitic solid solution. The dense occurrence of blocky tungsten carbide particles in the microstructure is reasonably related to the electrode filler composition (60 wt.% W_2_C). Thermally-induced cracks are clearly visible in the microstructure. They are presumably caused by volume changes of individual microstructural constituents during the solidification [26] and their formation may be a result of high cooling rate and a difference in thermal expansion coefficients of the carbide and the matrix [27]. Thanks to partial melting of W_2_C particle/matrix interfaces (seen more clearly on SEM image in Figure 3b), an original matrix solid solution is getting enriched by a certain amount of dissolved tungsten and carbon. This effect gives rise to the formation of additional precipitate phases, such as WC and M_6_C carbides which were indeed indicated by both thermodynamic calculation (Figure 3c) and XRD measurement (Figure 3d). Thermodynamic calculation was performed for estimated modification of the hardfacing matrix composition containing up to 15 wt.% W and 0.6 wt.% C, taking into account the considered W_2_C partial melting effects. The presence of thermodynamically predicted boride phase has not been experimentally evidenced by performed XRD measurement.

Figure 4a shows light-optical micrograph of dendritic microstructure of “LNM 420FM” hardfacing produced by MAG welding method using protective gas (82% Ar, 18% CO_2_). The detailed microstructure of the hardfacing is shown in Figure 4b. Thanks to the “LNM 420FM” hardfacing chemical composition (Table 2), including high chromium, silicon and carbon content (9 wt.% Cr, 3 wt.% Si, 0.5 wt.% C), the microstructure solidifies through high-temperature ferrite (i.e., δ-ferrite) region and then transforms to austenite phase (Figure 4c). According to the thermodynamic calculation (Figure 4c), the final solidified microstructure is formed of ferritic matrix with secondary precipitates of chromium carbide of M_7_C_3_ type. However, due to the presence of δ-ferrite and austenite in high temperature range and relatively high cooling rate during the hardfacing production, subsequent on-cooling formation of mixed ferritic/martensitic microstructure with M_7_C_3_ type carbides and some amount of retained austenite is reasonable. A corresponding XRD pattern of the hardfacing is shown in Figure 4d.

Figure 5a shows light-optical micrograph of dendritic microstructure of “E DUR 600” hardfacing produced by MMA welding method. Thanks to its chemical composition (Table 2), including high chromium, carbon, and tungsten content (8.5 wt.% Cr, 0.5 wt.% C, 0.5 wt.% W), the microstructure is very similar to the microstructure of previous “LNM 420FM” hardfacing. The present microstructure (Figure 5a,b) was also formed by solidification through high-temperature δ-ferrite region and then it transformed to austenite phase (Figure 5c). It also holds that due to the presence of δ-ferrite and austenite in high temperature range and high cooling rate during the hardfacing production, its final microstructure is formed of ferritic/martensitic matrix. The thermodynamic calculation also shows secondary phase precipitation of M_7_C_3_ and M_23_C_6_ chromium carbides. A corresponding XRD pattern of the hardfacing is shown in Figure 5d. It has indeed indicated the presence of ferritic/martensitic matrix with M_7_C_3_ type carbides and some amount of retained austenite.

Figure 6a shows a light-optical micrograph of dendritic microstructure of “Weartrode 62” hardfacing produced by MMA welding method. Thanks to its chemical composition (Table 2), including high chromium, silicon, titanium, vanadium and carbon content (6.3 wt.% Cr, 2 wt.% Si, 4.8 wt.% Ti, 5 wt.% V, 3 wt.% C), the microstructure contains complex titanium/vanadium carbides of MC type (see detail in Figure 6b), which precipitated primarily from the liquid phase and partially within residual eutectic. The present hardfacing solidified through the austenitic region (Figure 6c) which gave rise to the possibility of final ferritic/martensitic microstructure creation. According to the thermodynamic calculation (Figure 6c), the microstructure may also contain some small amount of secondary M_7_C_3_ chromium carbides. The XRD pattern of the hardfacing (Figure 6d) has also confirmed some amount of retained austenite in addition to other phases.

### 3.2. Hardness

Figure 7 shows a graphical comparison of the average surface hardness values corresponding to individual hardfacings and 16MnCr5 steel reference material.

It is clearly seen that individual hardfacings exhibit mutually differing surface hardness values which can be directly related to great diversity in their chemical composition (Table 2) and microstructural characteristics (Figure 2, Figure 3, Figure 4, Figure 5 and Figure 6). The highest average surface hardness exhibits “RD 571” hardfacing thanks to the presence of hard tungsten carbide particles (see Figure 3) in its microstructure. However, this hardfacing also shows the greatest scattering of hardness values due to significant differences between the hardness of very hard tungsten carbide particles and that of relatively soft austenitic matrix. The relatively high hardness values with low value scattering are observed for “E DUR 600” and “Weartrode 62” hardfacings, whereas “E520 RB” and “LNM 420FM” hardfacings show lower average hardness values at greater value scattering. Lower hardness of “E520 RB” hardfacing may be attributed to its non-hardenable δ-ferrite primary dendritic structure (Figure 2), whereas the lower hardness of “LNM 420FM” hardfacing is related to its significantly self-tempered martensitic/ferritic microstructure (Figure 4). In contrast, the high hardness of “E DUR 600” and “Weartrode 62” hardfacings can be attributed to the presence of fresh (i.e., non-tempered) martensite within their solidified microstructures (Figure 5 and Figure 6). Apart from this, the hardness of “E DUR 600” hardfacing is additionally enhanced by a solid solution strengthening effect from the used tungsten alloying (i.e., 0.5 wt.% W). The additional rise in hardness of the “Weartrode 62” hardfacing is reasonably caused by the presence of bulky titanium/vanadium carbides of MC type in its microstructure (Figure 6). In spite of the described differences in hardness characteristics of individual hardfacings, it can be stated that all investigated hardfacings exhibit notably increased hardness compared to 16MnCr5 grade steel. This result gives rise to the assumption of greater abrasive wear resistance of all the individual hardfacings than that of the reference steel material.

### 3.3. Abrasive Wear Resistance

Figure 8 represents a graphical interpretation of the average weight loss of individual investigated materials subjected to the specified three-body abrasive wear testing (Figure 1).

As expected, the highest weight loss, i.e., the lowest abrasive wear resistance was exhibited by the 16MnCr5 grade reference steel without hardfacing. On the other hand, all the used hardfacings exhibited lower weight loss compared to the reference steel. In order to estimate eventual correlation between abrasive wear resistance and hardness characteristics, Figure 9 shows the graphical dependence of relative abrasive wear resistance on the hardness ratio (i.e., the ratio of the tested material hardness to the used SiO_2_ abradant hardness).

It can be clearly seen from Figure 9 that no general correlation can be found between abrasive wear resistance of the investigated hardfacings and their hardness. This result can be explained in terms of strongly differing chemical compositions (Table 2) and microstructures (Figure 2, Figure 3, Figure 4, Figure 5 and Figure 6) of individual hardfacings. However, the obtained results of the present study clearly indicate that the governing factor enhancing abrasive wear resistance is strongly related to overall carbon content of the used hardfacing material. Thus, in the conditions of the present investigation, the high carbon or high carbide containing hardfacings, namely “E520 RB” (3.5 wt.% C), “RD 571” (60 wt.% W_2_C), and “Weartrode 62” (3 wt.% C), exhibited very high abrasive wear resistance compared to other studied hardfacings (“LNM 420FM” and “E DUR 600”) with much lower carbon content (0.5 wt.% C). Finally, it has been concluded that the “E520 RB” hardfacing exhibited the highest abrasive wear resistance thanks to its appropriate hardness and beneficial “ledeburite-type” eutectic microstructure.

### 3.4. Characteristics of Wear Tracks and Wear Mechanisms

Figure 10 shows cross-sectional profiles of wear tracks generated on the surfaces of individual samples after the performed abrasion wear tests.

From these wear track profiles the information about the reached depth of penetration and wear track morphology can be obtained. Among all the tested materials, the reference (16MnCr5 grade steel) material with the lowest abrasive wear resistance (Figure 8), exhibits the highest penetration depth and the smoothest wear track pattern (Figure 10a). On the contrary, all the hardfacings with higher abrasive wear resistance, show lower penetration depths and a recognizably roughened wear track pattern (Figure 10b–f). As expected, the hardfacings with the highest abrasive wear resistance, namely “E520 RB”, “RD 571”, and “Weartrode 62” show the lowest penetration depths and significantly roughened wear track patterns (Figure 10b,c,f), whereas the hardfacings with poorer abrasive wear resistance, i.e., “LNM 420FM” and “E DUR 600” show higher penetration depths and relatively smoother wear track patterns (Figure 10d,e). It should be noted that the extremely irregular morphology of wear track pattern of the “RD 571” hardfacing can be explained in terms of its extraordinary heterogeneous “cermet-type” microstructure (Figure 3a). Thus the wear track pattern of “RD 571” hardfacing (Figure 10c) is a result of highly inhomogeneous wear process including easy wear of soft austenitic matrix and significantly suppressed wear of bulky tungsten carbide particles. However, although the “RD 571” hardfacing exhibited quite satisfactory abrasive wear resistance in laboratory testing conditions (Figure 8), its suitability for practical application in heavy-duty forestry operation is assumed to be not very good. This opinion can likely be supported by the fact of more complex loading conditions during the chipper tool service, including not only abrasion wear but also irregular impact and fatigue degradation. For such a complex loading, the presence of bulky carbides in microstructure may be considered rather unfavorable. Similarly, the “Weartrode 62” hardfacing also contains in its microstructure significant amount of bulky titanium/vanadium carbides (Figure 6b) which also may act like crack nucleation sites in forestry service conditions. In contrast, the “E520 RB” hardfacing, thanks to its coarse “ledeburite-type” microstructure with specific “interlocking” eutectic morphology (Figure 2b), exhibits among the studied hardfacings the highest abrasive wear resistance (Figure 8) in laboratory testing conditions. Thus the “E520 RB” hardfacing can likely be considered the best candidate for the application on heavy-duty chipper tools in forestry maintenance operations. The resisting behavior of this material against abrasive wear is clearly documented in Figure 10b by significant serrations of the wear track pattern. Figure 11 demonstrates direct proportionality between average weight loss and maximal wear track depth of investigated materials.

A slight deviation from the demonstrated correlation is observed for the “RD 571” hardfacing which exhibits, by its greater value of maximal wear track depth, slightly lower average weight loss compared to the “Weartrode 62” hardfacing with lower value of maximal wear track depth. However, this observation can be easily explained by the occurrence of strongly irregular morphology of cross-sectional wear track profile of “RD 571” hardfacing (Figure 10c) which has been already interpreted and discussed.

In Figure 12, typical wear mechanisms are shown for the investigated hardfacings exhibiting significant differences in their microstructural features influencing the resulting abrasive wear resistance. Figure 12a shows the wear track topography for the “E520 RB” hardfacing with the highest abrasive wear resistance. As expected, the observed wear track characteristics do not indicate any severe wear damage mechanisms. As mentioned earlier, this behavior can likely be related to the beneficial “interlocking” effect of the “ledeburite-type” eutectic microstructure (Figure 2b). The wear track topographies of the “Weartrode 62” and “RD 571” hardfacings with slightly worse abrasive wear resistance than the “E520 RB” hardfacing are shown in Figure 12b,c, respectively. From the observation of the wear track topography of the “Weartrode 62” hardfacing (Figure 12b), it can be clearly indicated that abrasive wear mechanisms are limited to some isolated cutting and delamination effects in the vicinity of the precipitate particles. On the other hand, in Figure 12c, showing the wear track surface of the “RD 571” hardfacing, the acting wear mechanisms are related to intensive plastic deformation and preferential removal of the austenitic steel matrix accompanied by retarded surface abrasion of bulky tungsten carbide particles leading to the overall wear suppression. Finally, Figure 12d shows the wear track topography of the “LNM 420FM” hardfacing with the lowest abrasive wear resistance. The wear track pattern clearly indicates the wear mechanism to be pronounced cutting which is generally considered to be the most severe abrasion wear mechanism. It is generally accepted that the cutting mechanism results in the most intensive removal of the abraded material from its surface. Thus, the observed cutting mechanism correlates well with the lowest abrasive wear resistance of the “LNM 420FM” hardfacing.

The performed topographic analyses of the wear tracks on individual studied hardfacings enabled the correlation of their abrasive wear resistance with observed wear mechanisms acting during the abrasion wear tests in laboratory testing conditions. However, experimental testing of individual hardfacings on real chipper tools in forestry service conditions is necessary to examine practical validity of research findings obtained in the present investigation.

## 4. Summary and Conclusions

The present study was focused on laboratory investigation of microstructure and abrasive wear resistance of five different alloy hardfacings on 16MnCr5 grade low carbon steel for application on heavy-duty chipper tools in forestry maintenance operations. The obtained results are summarized in the following conclusions:
Microstructural and phase analyses of individual studied hardfacings revealed great diversity in their microstructures and phase composition, which can be primarily related to their chemical composition variation. The high chromium and carbon contents (25 wt.% Cr, 3.5 wt.% C) in “E520 RB” hardfacing resulted in a formation of coarse “ledeburite-type” microstructure with primary δ-ferrite dendrites and residual eutectics with Cr-rich carbides. Thanks to the high nickel content (15 wt.% Ni) and W_2_C filling powder (60 wt.% W_2_C) of the used cored electrode, the “RD 571” hardfacing exhibited highly heterogeneous “cermet-type” microstructure with austenitic matrix and bulky tungsten carbide particles. Due to high chromium content in “LNM 420FM” (9 wt.% Cr), “E DUR 600” (8.5 wt.% Cr) and “Weartrode 62” (6.3 wt.% Cr) Fe-based hardfacings, their microstructures were all found to be formed of dendritic cell structures containing ferrite, martensite, retained austenite, and various carbide precipitates (e.g., Cr-rich carbides, Ti, V-rich carbides), depending on the individual hardfacing chemical composition.Thanks to varying microstructural heterogeneity of individual hardfacings, their surface hardness values were also mutually differing. All the studied hardfacings exhibited notably increased hardness values compared to 16MnCr5 grade steel reference material. The relatively high average hardness values at low value scattering were determined for “E DUR 600” and “Weartrode 62” hardfacings, probably due to the presence of fresh martensite within their solidified microstructures. Additional hardening might also occur as a result of solid solution strengthening by tungsten (0.5 wt.% W) and precipitation strengthening by (Ti, V)C carbides in “E DUR 600” and “Weartrode 62” hardfacings, respectively. In contrast, the “E520 RB” and “LNM 420FM” hardfacings exhibited considerably lower hardness values at greater value scattering that might be attributed to the presence of relatively softer ferrite matrix and harder Cr-rich carbides in their microstructures. The “RD 571” hardfacing exhibited the highest hardness but extraordinary great value scattering due to significant differences between the hardness values of hard tungsten carbide particles and those of much softer austenitic matrix.All the investigated hardfacings, i.e., “E520 RB”, “RD 571”, “LNM 420FM”, “E DUR 600”, and “Weartrode 62”, exhibited lower weight loss and thus higher abrasive wear resistance, compared to the 16MnCr5 grade reference steel material. Due to vastly differing microstructure types (e.g., “ledeburite-type” with primary ferrite dendrites and chromium carbide eutectic, “cermet-type” with austenitic matrix and various tungsten carbides, mixed dendritic structures with ferritic/martensitic matrix and various carbide precipitates) of individual hardfacings, a general correlation between their surface hardness and abrasive wear resistance has not been indicated. However, the results of the present investigation obviously indicated the crucial factor enhancing the abrasive wear resistance to be the overall carbon content of the hardfacing material. Thus, the high abrasive wear resistance exhibited the hardfacings “E520 RB” (3.5 wt.% C), “RD 571” (60 wt.% W_2_C), and “Weartrode 62” (3 wt.% C), compared to other studied hardfacings (“LNM 420FM” and “E DUR 600”) with much lower carbon content (0.5 wt.% C). Among these hardfacings, the “E520 RB” material exhibited the highest abrasive wear resistance thanks to its beneficial “ledeburite-type” microstructure with specific “interlocking” eutectic morphology. The morphological wear track analyses revealed typical features of governing abrasive wear mechanisms of individual hardfacings. Despite the obtained results, service testing of individual hardfacings on real chipper tools is necessary to correlate currently obtained research findings with abrasive wear behavior in forestry operation conditions.


## Figures and Tables

**Figure 1 materials-12-02212-f001:**
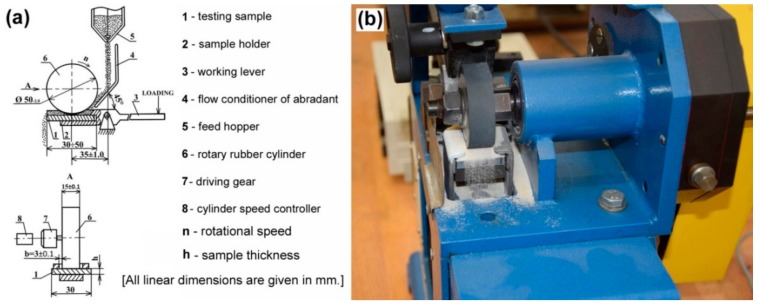
Testing apparatus for three-body abrasive wear test according to “ГОСТ 23.208-79” standard [25]: schematic layout (**a**) and detailed photograph of the apparatus at work (**b**).

**Figure 2 materials-12-02212-f002:**
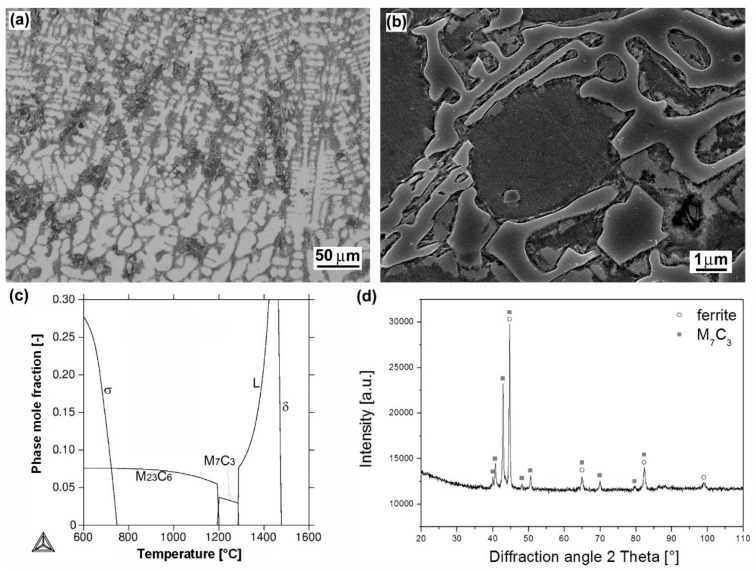
Characterization of “E520 RB” hardfacing: overall microstructure visualized by light-optical microscope (LOM) (**a**); detailed microstructure visualized by scanning electron microscope (SEM) (**b**); thermodynamic calculation of equilibrium phase composition depending on temperature (**c**); X-ray diffraction (XRD) pattern representing experimentally determined phase composition (**d**).

**Figure 3 materials-12-02212-f003:**
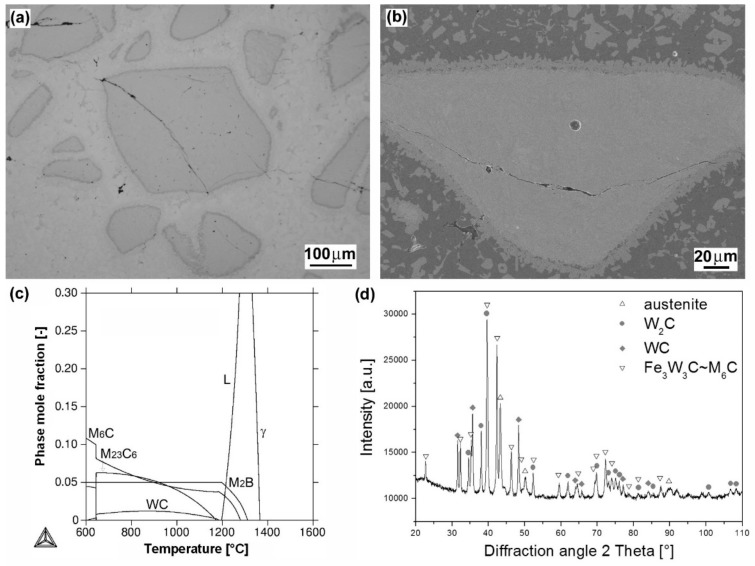
Characterization of “RD 571” hardfacing: overall microstructure visualized by LOM (**a**); detailed microstructure visualized by SEM (**b**); thermodynamic calculation of equilibrium phase composition depending on temperature (**c**); XRD-pattern representing experimentally determined phase composition (**d**).

**Figure 4 materials-12-02212-f004:**
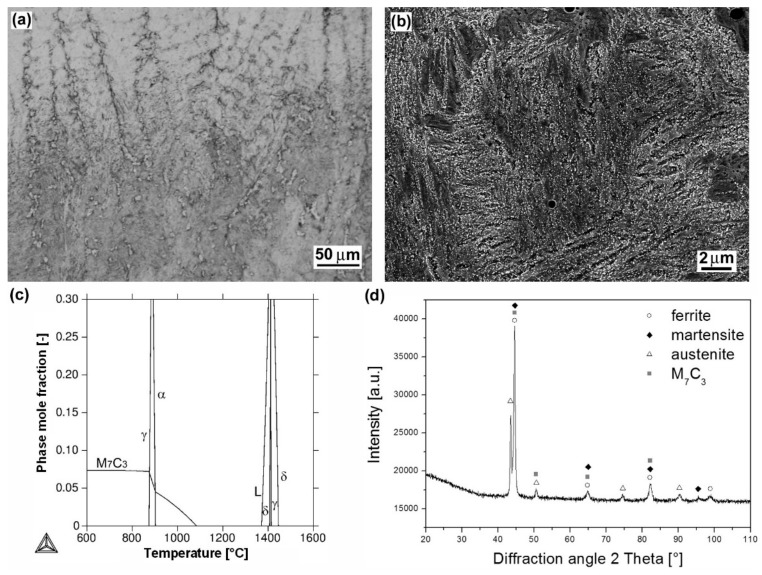
Characterization of “LNM 420FM” hardfacing: overall microstructure visualized by LOM (**a**); detailed microstructure visualized by SEM (**b**); thermodynamic calculation of equilibrium phase composition depending on temperature (**c**); XRD-pattern representing experimentally determined phase composition (**d**).

**Figure 5 materials-12-02212-f005:**
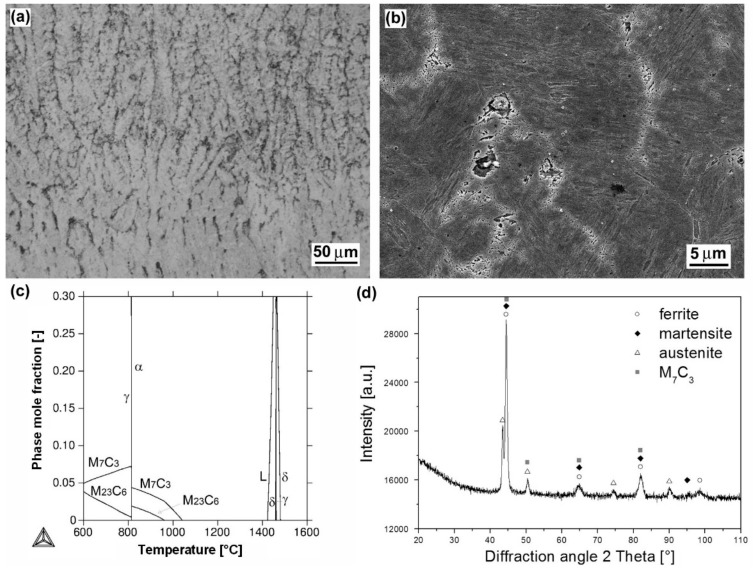
Characterization of “E DUR 600” hardfacing: overall microstructure visualized by LOM (**a**); detailed microstructure visualized by SEM (**b**); thermodynamic calculation of equilibrium phase composition depending on temperature (**c**); XRD-pattern representing experimentally determined phase composition (**d**).

**Figure 6 materials-12-02212-f006:**
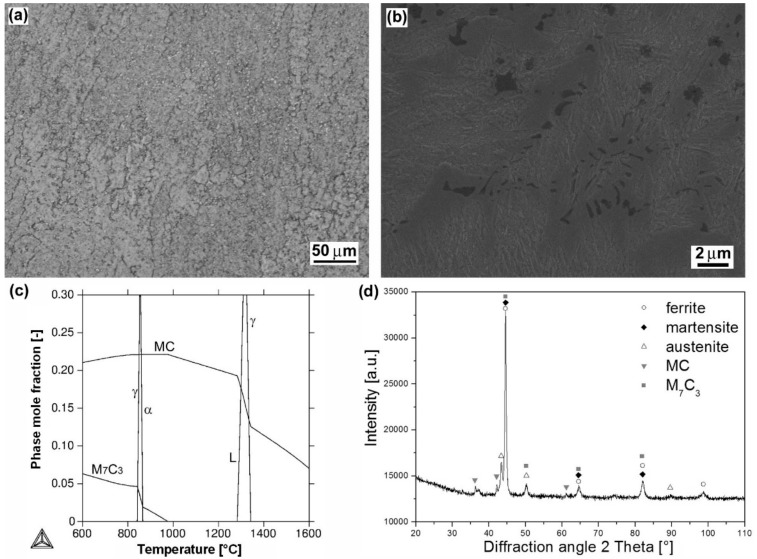
Characterization of “Weartrode 62” hardfacing: overall microstructure visualized by LOM (**a**); detailed microstructure visualized by SEM (**b**); thermodynamic calculation of phase composition depending on temperature (**c**); XRD-pattern representing experimentally determined phase composition (**d**).

**Figure 7 materials-12-02212-f007:**
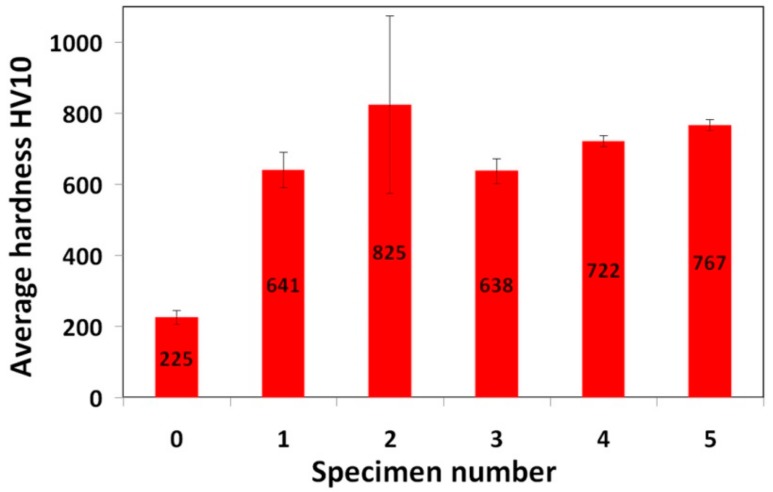
Average surface hardness values of investigated materials: 0—16MnCr5 steel; 1—“E520 RB” hardfacing; 2—“RD 571” hardfacing; 3—“LNM 420FM” hardfacing; 4—“E DUR 600” hardfacing; 5—“Weartrode 62” hardfacing.

**Figure 8 materials-12-02212-f008:**
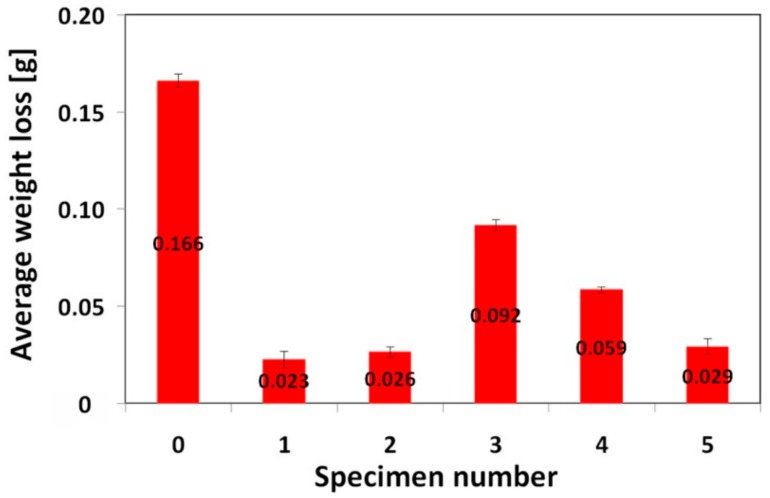
Average weight loss of investigated materials subjected to abrasive wear testing: 0—16MnCr5 steel; 1—“E520 RB” hardfacing; 2—“RD 571” hardfacing; 3—“LNM 420FM” hardfacing; 4—“E DUR 600” hardfacing; 5—“Weartrode 62” hardfacing.

**Figure 9 materials-12-02212-f009:**
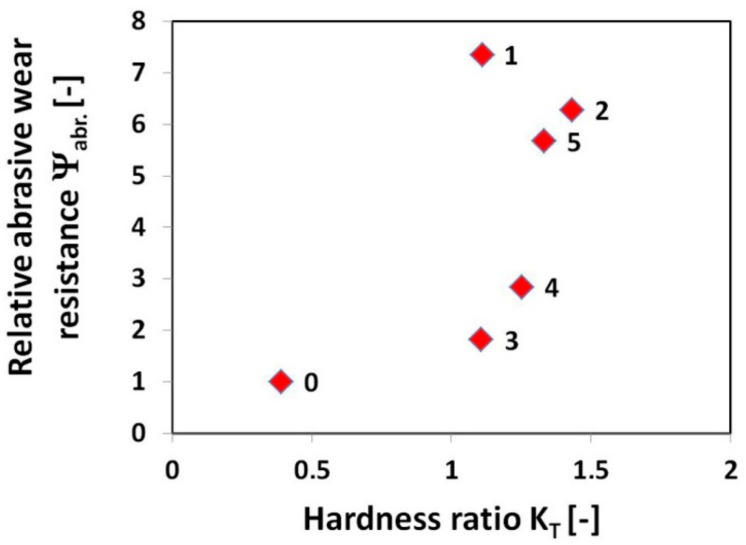
Relative abrasive wear resistance in relation to hardness ratio: 0—16MnCr5 steel; 1—“E520 RB” hardfacing; 2—“RD 571” hardfacing; 3—“LNM 420FM” hardfacing; 4—“E DUR 600” hardfacing; 5—“Weartrode 62” hardfacing.

**Figure 10 materials-12-02212-f010:**
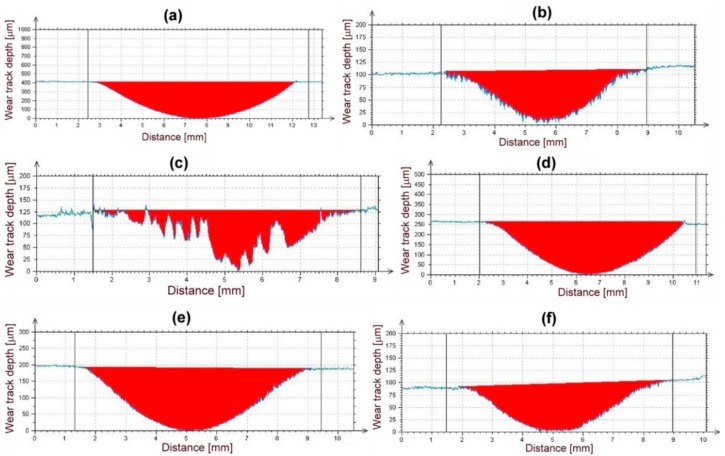
Typical cross-sectional profiles of wear tracks generated on the surfaces of investigated materials after abrasion wear tests: 16MnCr5 grade steel (**a**); “E520 RB” hardfacing (**b**); “RD 571” hardfacing (**c**); “LNM 420FM” hardfacing (**d**); “E DUR 600” hardfacing (**e**); “Weartrode 62” hardfacing (**f**).

**Figure 11 materials-12-02212-f011:**
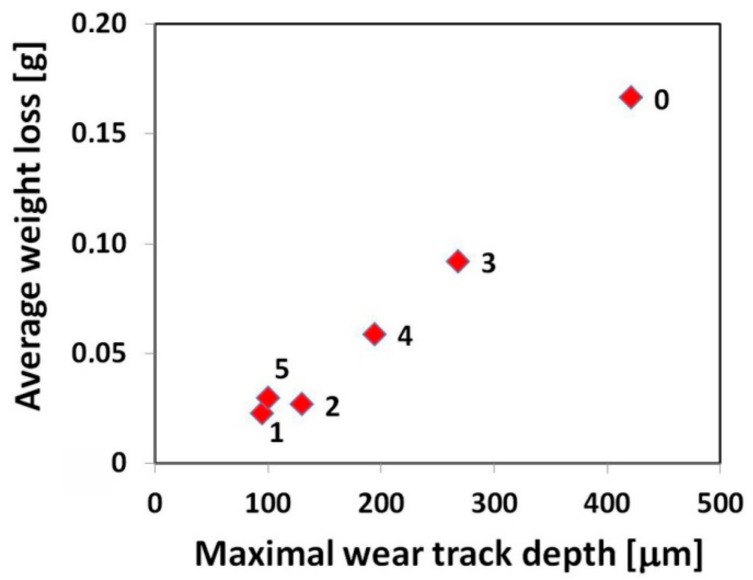
Correlation between average weight loss and maximal wear track depth of investigated materials after abrasion wear testing: 0—16MnCr5 steel; 1—“E520 RB” hardfacing; 2—“RD 571” hardfacing; 3—“LNM 420FM” hardfacing; 4—“E DUR 600” hardfacing; 5—“Weartrode 62” hardfacing.

**Figure 12 materials-12-02212-f012:**
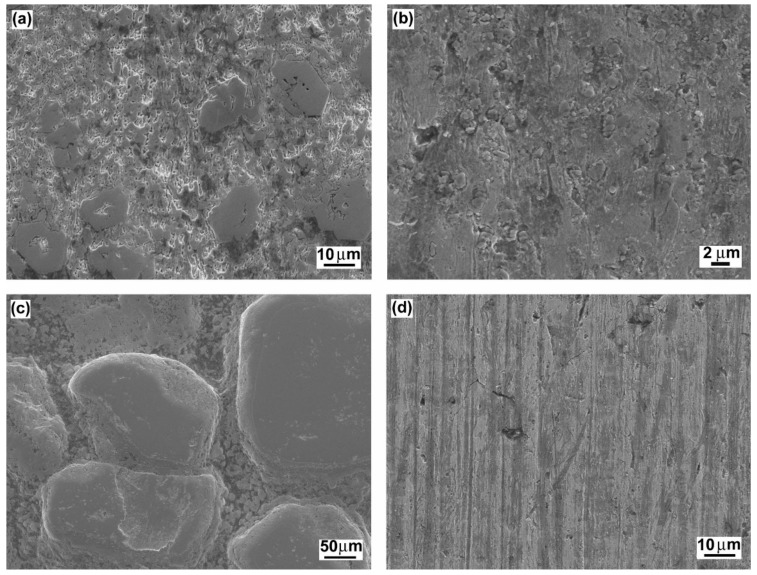
Typical wear mechanisms of selected hardfacings with significantly differing abrasive wear resistance: “E520 RB” (**a**); “Weartrode 62” (**b**); “RD 571” (**c**); “LNM 420FM” (**d**).

**Table 1 materials-12-02212-t001:** Chemical composition (wt.%) of 16MnCr5 grade steel reference material.

**C**	**Mn**	**Si**	**P**	**S**	**Al**	**Cu**	**Ni**	**Cr**	**V**
0.212	1.302	0.241	0.012	0.006	0.028	0.125	0.131	1.219	0.004
**Nb**	**Mo**	**Co**	**Sn**	**Sb**	**W**	**B**	**Ca**	**Zr**	**Fe**
<0.002	0.035	0.009	0.007	0.003	<0.003	0.0002	0.0017	0.001	rest

**Table 2 materials-12-02212-t002:** Chemical composition (wt.%) of input materials for realization of hard surfacing.

Specimen No.	Hardfacing Material	C	Mn	Si	Ni	Cr	Ti	V	W	B	W_2_C	Fe
**1**	**E520 RB**	3.50	0.80	0.80		25		1.3				rest
**2**	**RD 571**	0.10	0.50	0.80	15.0	5.0				0.3	60	rest
**3**	**LNM 420FM**	0.50	0.40	3.00		9.0						rest
**4**	**E DUR 600**	0.50				8.5			0.5			rest
**5**	**Weartrode 62**	3.00	0.30	2.00		6.3	4.8	5.0				rest

**Table 3 materials-12-02212-t003:** Welding methods and process parameters used for fabrication of the hardfacings.

Parameter	Hardfacing Materials Used for Hard Surfacing of 16MnCr5 Grade Steel Reference Material				
	E520 RB	RD 571	LNM 420FM	E DUR 600	Weartrode 62
Electrode dimensions	ϕ 2.5 × 450 mm	ϕ 3.2 × 450 mm	ϕ 1.2 mm	ϕ 2.5 × 450 mm	ϕ 2.5 × 350 mm
Welding current/voltage	120–130 A 24 V	-	180 A 28.8 V	80 A 22.5 V	109 A 19.2 V
Welding speed (cm·min^−1^)	15	-	27–32	21–29	21–29
Preheat condition	-	-	150–180 °C	150–180 °C	150–180 °C
Welding method	MMA (111)	Oxy-acetylene flame-softly carburizing	MAG (135), M21: 82%Ar, 18%CO_2_	MMA (111)	MMA (111)

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
