# Peer review of "Microstructure and Abrasive Wear Resistance of Various Alloy Hardfacings for Application on Heavy-Duty Chipper Tools in Forestry Shredding and Mulching Operations"

_materials, 2019, doi:10.3390/ma12132212_

Round 1
Reviewer 1 Report
The aim of the paper, i.e. the abrasive wear resistance of commercially available five different alloy hardfacings on 16MnCr5 grade low-carbon ferritic-pearlitic steel, is quite interesting, even if not so appealing, due to the large number of papers already published on similar process and materials. Authors analyzed the results of abrasive wear tests of the investigated hardfacings and discussed them in relation to their microstructure, hardness, and wear mechanisms characteristics.
The abstract summarize the work. The purpose of the study is clearly outlined and the findings of prior work are well discussed. There are no errors in logic or experimental procedure. The authors accurately explain how the data were collected. There is sufficient information that the experiment can be reproduced. All topics are well presented and discussed. The summary and conclusions are sound and justified. All presented figures are good quality and they prove their point. The paper is written in good English. The manuscript is easily readable concerning language, style and presentation. The references are appropriate and up to date.
Author Response
Detailed response to Reviewer 1
on his/her comments on the manuscript (ID: materials-545651) “Microstructure and Abrasive Wear Resistance of Various Alloy Hardfacings for Application on Heavy-Duty Chipper Tools in Forestry Shredding and Mulching Operations” by authors Ladislav Falat, Miroslav DĹľupon, Miroslava Ťavodová, Richard Hnilica, Veronika Ä˝uptáčiková, Lucia ÄŚiripová, Viera Homolová, KatarĂna ÄŽurišinová, submitted to journal „MATERIALS (by MDPI)”.
Košice, Slovakia, July 7th, 2019
Dear Reviewer 1,
Thank you very much for your careful review and positive comments approving our manuscript. Taking into account your comments as well as recommendations of two other reviewers; I have now submitted a revised version of our manuscript for its final evaluation. All the performed modifications in the revised manuscript are clearly highlighted in the manuscript text using red-colored font. The detailed response to your comments is provided below:
Reviewer 1, comment 1:
The aim of the paper, i.e. the abrasive wear resistance of commercially available five different alloy hardfacings on 16MnCr5 grade low-carbon ferritic-pearlitic steel, is quite interesting, even if not so appealing, due to the large number of papers already published on similar process and materials. Authors analyzed the results of abrasive wear tests of the investigated hardfacings and discussed them in relation to their microstructure, hardness, and wear mechanisms characteristics.
Our response:
Thank you very much for your approval. Just as a little response: It is really true that there are plenty of papers dealing with similar processes and materials, e.g. focusing on tools used in agriculture, road building, and mining industry. However, the degradation behavior of forestry tools has not been up to now so widely investigated which may be the main reason for pertaining problems concerning premature failures of forestry tools. We believe that our present contribution might be a challenge to wider research community to address this topic on a larger scale.
Reviewer 1, comment 2:
The abstract summarize the work. The purpose of the study is clearly outlined and the findings of prior work are well discussed. There are no errors in logic or experimental procedure. The authors accurately explain how the data were collected. There is sufficient information that the experiment can be reproduced. All topics are well presented and discussed. The summary and conclusions are sound and justified. All presented figures are good quality and they prove their point. The paper is written in good English. The manuscript is easily readable concerning language, style and presentation. The references are appropriate and up to date.
Our response:
Thank you very much for your commentary approving our manuscript for the journal publication.
With kind regards
Ladislav Falat
Corresponding author
Dr. Ladislav Falat
Institute of Materials Research
Slovak Academy of Sciences
Watsonova 47, 040 01 Košice
Slovakia
E-mail: [email protected]

Reviewer 2 Report
This is an interesting and well written work that fits well within the scope of this Journal. I only have very few comments that require some minor clarifications. These are given hereafter:
1. To identify the carbide phases, you used thermodynamic calculations. However, welding processes are non-equilibrium. How can you be sure of the type of carbides that you are forming?
2. Did you measure the size of carbide particles? As you correctly mention in the discussion the size and distribution of hard particles in a material can have a significant effect on its tribological performance.
3. Material LNM 420FM, despite having high hardness fluctuation (which relates to its structural inhomogeneity) and an ununiform wear track, the experimental fluctuation of its average weight loss is rather small and comparable to the other materials.
Author Response
Detailed response to Reviewer 2
on his/her comments on the manuscript (ID: materials-545651) “Microstructure and Abrasive Wear Resistance of Various Alloy Hardfacings for Application on Heavy-Duty Chipper Tools in Forestry Shredding and Mulching Operations” by authors Ladislav Falat, Miroslav DĹľupon, Miroslava Ťavodová, Richard Hnilica, Veronika Ä˝uptáčiková, Lucia ÄŚiripová, Viera Homolová, KatarĂna ÄŽurišinová, submitted to journal „MATERIALS (by MDPI)”.
Košice, Slovakia, July 7th, 2019
Dear Reviewer 2,
Thank you very much for your careful review and positive comments approving our manuscript. Taking into account your comments as well as recommendations of two other reviewers; I have now submitted a revised version of our manuscript for its final evaluation. All the performed modifications in the revised manuscript are clearly highlighted in the manuscript text using red-colored font. The detailed response to your comments is provided below:
Reviewer 2, comment 1:
This is an interesting and well written work that fits well within the scope of this Journal. I only have very few comments that require some minor clarifications. These are given hereafter:
1. To identify the carbide phases, you used thermodynamic calculations. However, welding processes are non-equilibrium. How can you be sure of the type of carbides that you are forming?
Our response:
Thank you very much for this important notice. As it has already been noticed in our original manuscript, thermodynamic equilibrium calculations were performed for our studied hardfacings in order to predict their phase composition which might be theoretically expected. Moreover, thermodynamic calculations were very useful to estimate solidification behavior of the hardfacings (e.g., primary solidified phase, nature of precipitation reactions, etc.). The real phase composition of the studied non-equilibrium hardfacings could only be determined experimentally by means of XRD analyses. For real microstructure formation; kinetic factor is very important besides thermodynamic factor. Thus, we have also pointed out that during the hardfacings preparation, some phases might occur whereas the others not. Due to fast diffusion kinetics of carbon in all studied hardfacings during their solidification, some of predicted carbide precipitates can be reasonably expected. In contrast, the occurrence of thermodynamically predicted intermetallic phases in real hardfacings microstructures is rather unlikely. The equilibrium calculations also could not predict martensite phase. This could only be justified by XRD analyses and microstructural observations.
Reviewer 2, comment 2:
2. Did you measure the size of carbide particles? As you correctly mention in the discussion the size and distribution of hard particles in a material can have a significant effect on its tribological performance.
Our response:
Thank you for this remark. Size and distribution of hard particles is one of crucial factors influencing abrasive wear resistance. However, our present pilot study about the five selected hardfacings was rather focused on principal differentiation of individual microstructure types and their chemical and phase compositions in relation to the resulting properties. The detailed quantitative characterization of particle size distributions in individual hardfacings will be the subject of our subsequent studies.
Reviewer 2, comment 3:
3. Material LNM 420FM, despite having high hardness fluctuation (which relates to its structural inhomogeneity) and an ununiform wear track, the experimental fluctuation of its average weight loss is rather small and comparable to the other materials.
Our response:
Thank you very much for this interesting notice. It is really true that regardless of hardness values fluctuations, all the studied hardfacings exhibit rather small fluctuations around the values of their average weight losses. At present point of investigation, we believe that this behavior is due to the fact that the weight loss represents an integral quantity. In principle, it depends on the used wear test conditions. It seems that under conditions of present laboratory investigation, the used testing conditions have assured such observed uniform behavior thanks to relatively high pressure force and long sliding distance during abrasive wear test. In our future investigations, we also plan to study abrasive wear behavior depending on changing wear test conditions in order to clarify possible controlling effects in more detail.
Thank you once again very much for all your valuable comments. We hope our responses will meet with your approval.
With kind regards
Ladislav Falat
Corresponding author
Dr. Ladislav Falat
Institute of Materials Research
Slovak Academy of Sciences
Watsonova 47, 040 01 Košice
Slovakia
E-mail: [email protected]

Reviewer 3 Report
The manuscript is focused on the investigation of abrasive wear resistance of five hardfacings on 16MnCr5 grade low-carbon ferritic-pearlitic steel in laboratory conditions.
Paper is well prepared and I would like to recommend only minor corrections before accepting this work for publication at Materials.
There is no information about the reason for a particular selection of five hardfacings for the present work.
I recommend to include in Fig. 10 legend for materials for each from (a)-(f) figures, at least in way 0-5 as it is used in other Figures or better – full name of each hardfacings. In my own opinion, it is better to use full names, such as “E520 RB”, “RD 571”, “LNM 420FM” etc., in all figures. However, I understand that in Fig. 9 it is not convenient.
There are few misprints in row 332-334, while referencing to Fig.10 a-f. Final double check is essential.
“Summary and conclusions” can be simplified and shorten. Only the main conclusions should be highlighted, more details can be found in the body of the manuscript.
Generally, after minor revision, I recommend acceptance of the manuscript for publication in Materials.
Author Response
Detailed response to Reviewer 3
on his/her comments on the manuscript (ID: materials-545651) “Microstructure and Abrasive Wear Resistance of Various Alloy Hardfacings for Application on Heavy-Duty Chipper Tools in Forestry Shredding and Mulching Operations” by authors Ladislav Falat, Miroslav DĹľupon, Miroslava Ťavodová, Richard Hnilica, Veronika Ä˝uptáčiková, Lucia ÄŚiripová, Viera Homolová, KatarĂna ÄŽurišinová, submitted to journal „MATERIALS (by MDPI)”.
Košice, Slovakia, July 7th, 2019
Dear Reviewer 3,
Thank you very much for your careful review and positive comments approving our manuscript. Taking into account your comments as well as recommendations of two other reviewers; I have now submitted a revised version of our manuscript for its final evaluation. All the performed modifications in the revised manuscript are clearly highlighted in the manuscript text using red-colored font. The detailed response to your comments is provided below:
Reviewer 3, comment 1:
The manuscript is focused on the investigation of abrasive wear resistance of five hardfacings on 16MnCr5 grade low-carbon ferritic-pearlitic steel in laboratory conditions. Paper is well prepared and I would like to recommend only minor corrections before accepting this work for publication at Materials. There is no information about the reason for a particular selection of five hardfacings for the present work.
Our response:
Thank you for your remark. In our original manuscript, the following information has been mentioned: “However, there is no unified opinion among researchers on the most beneficial type of material microstructure in terms of its abrasive wear resistance. Depending on specific application, either martensitic-carbidic or austenitic-carbidic microstructures are considered to be the most suitable for abrasive wear performance [22-24]. These different views arise from the fact of great diversity in abrasive wear processes as well as wide range of tool operational conditions. Therefore, it is highly important that materials investigations aimed at improvement of abrasive wear resistance of specific tools or structures have always to take into account the mentioned considerations.” In addition, the following text was added into the revised manuscript: “The selection of particular hardfacing materials was based on the assumption of creation of substantially differing microstructures and phase compositions of individual hardfacings in order to perform their pilot comparative study in terms of their abrasive wear resistance in laboratory testing conditions.” (Please see the lines 110-113 in the revised manuscript).
Reviewer 3, comment 2:
I recommend to include in Fig. 10 legend for materials for each from (a)-(f) figures, at least in way 0-5 as it is used in other Figures or better – full name of each hardfacings. In my own opinion, it is better to use full names, such as “E520 RB”, “RD 571”, “LNM 420FM” etc., in all figures. However, I understand that in Fig. 9 it is not convenient.
Our response:
Thank you very much for your kind recommendation. However, we are kindly asking you if we could let our original graphical arrangement of Figure 10 as well as that of the Figures 7-11. We believe that the original detailed description within individual figure captions will be as clear as accurate for the readership of the article.
Reviewer 3, comment 3:
There are few misprints in row 332-334, while referencing to Fig.10 a-f. Final double check is essential.
Our response:
Thank you very much for this important notice! We have made appropriate corrections in our revised manuscript: “As expected, the hardfacings with the highest abrasive wear resistance, namely “E520 RB”, “RD 571”, and “Weartrode 62” show the lowest penetration depths and significantly roughened wear track patterns (Figure 10b,c,f), whereas the hardfacings with poorer abrasive wear resistance, i.e., “LNM 420FM” and “E DUR 600” show higher penetration depths and relatively smoother wear track patterns (Figure 10d,e).” (Please note that the line numbering has changed in the revised manuscript due to new text addition according to your comment 1 above.)
Reviewer 3, comment 4:
“Summary and conclusions” can be simplified and shorten. Only the main conclusions should be highlighted, more details can be found in the body of the manuscript. Generally, after minor revision, I recommend acceptance of the manuscript for publication in Materials.
Our response:
Thank you for your opinion. However, we are kindly asking you if we could let our original conclusions within our revised manuscript. The conclusions were made as brief as possible to summarize the most important findings of the performed study. In our opinion, any shortening of the original conclusions would weaken the final impact of our presentation.
Thank you once again very much for all your valuable comments leading to our manuscript improvement. We hope our responses will meet with your approval.
With kind regards
Ladislav Falat
Corresponding author
Dr. Ladislav Falat
Institute of Materials Research
Slovak Academy of Sciences
Watsonova 47, 040 01 Košice
Slovakia
E-mail: [email protected]
